# Long term follow up of direct oral anticoagulants and warfarin therapy on stroke, with all-cause mortality as a competing risk, in people with atrial fibrillation: Sentinel network database study

Simon de Lusignan[1,2]*, F. D. Richard Hobbs[1]*, Harshana Liyanage[1], Julian Sherlock[1], Filipa Ferreira[1], Manasa Tripathy[1], Christian Heiss[3], Michael Feher[1], Mark P. Joy[1]

**1** Nuffield Department of Primary Care Health Sciences, University of Oxford, Oxford, United Kingdom, **2** Royal College of General Practitioners, London, United Kingdom, **3** Faculty of Health and Medical Sciences, University of Surrey, Guildford, United Kingdom

\* simon.delusignan@phc.ox.ac.uk (SL); richard.hobbs@phc.ox.ac.uk (FDRH)

**Data Availability Statement:** The RCGP RSC data set can be accessed by researchers, approval is on

## Abstract

### Background

We investigated differences in risk of stroke, with all-cause mortality as a competing risk, in people newly diagnosed with atrial fibrillation (AF) who were commenced on either direct oral anticoagulants (DOACs) or warfarin treatment.

### Methods and results

We conducted a retrospective cohort study of the Oxford Royal College of General Practitioners (RCGP) Research and Surveillance Centre (RSC) database (a network of 500 English general practices). We compared long term exposure to DOAC (n = 5,168) and warfarin (n = 7,451) in new cases of AF not previously treated with oral anticoagulants. Analyses included: survival analysis, estimating cause specific hazard ratios (CSHR), Fine-Gray analysis for factors affecting cumulative incidence of events occurring over time and a cumulative risk regression with time varying effects. We found no difference in CSHR between stroke 1.08 (0.72–1.63, p = 0.69) and all-cause mortality 0.93 (0.81–1.08, p = 0.37), or between the anticoagulant groups. Fine-Gray analysis produced similar results 1.07 (0.71–1.6 p = 0.75) for stroke and 0.93 (0.8–1.07, p = 0.3) mortality. The cumulative risk of mortality with DOAC was significantly elevated in early follow-up (67 days), with cumulative risk decreasing until 1,537 days and all-cause mortality risk significantly decreased coefficient estimate:: -0.23 (-0.38–0.01, p = 0.001); which persisted over seven years of follow-up.

### Conclusions

In this large, contemporary, real world primary care study with longer follow-up, we found no overall difference in the hazard of stroke between warfarin and DOAC treatment for AF.

a project-by-project basis (www.rcgp.org.uk/rsc).
Ethical approval by an NHS Research Ethics
Committee is needed before any data release/other
appropriate approval. If needed, requests for data
access from researchers who meet the criteria to
access confidential data can also be sent to orchid-
reg@phc.ox.ac.uk.

**Funding:** This work was funded by an
unconditional grant from Daiichi Sankyo (grant
number DSJP 3700) awarded to SdeL through his
university. This study was also funded by the
National Institute for Health Research (NIHR)
School for Primary Care Research, the NIHR
Collaboration for Leadership in Health Research
and Care (CLARHC) Oxford (grant number IS-SPC-
0514-10043), the NIHR Oxford Biomedical
Research Centre (BRC, UHT) (grant number IS-
BRC-1215-20005), and the NIHR Oxford Medtech
and In-Vitro Diagnostics Co-operative (MIC) (grant
number MIC-2016-018), all warded to FDRH. The
funders had no role in study design, data collection
and analysis, decision to publish, or preparation of
the manuscript.

**Competing interests:** The authors have read the
journal's policy and have the following competing
interests: Simon de Lusignan is the Director of the
Oxford RCGP RSC and has received funding
through his University for studies from Eli Lilly,
Astra-Zeneca, Sanofi, GSK, Seqirus and Takeda;
and been member of advisory boards for influenza
for Seqirus and Sanofi. FDRH has received
occasional fees from Bayer and Boehringer
Ingelheim for speaking or consulting on atrial
fibrillation related stroke risk. All other authors have
declared no financial relationships with any
organisations that might have an interest in the
submitted work in the previous three years, no
other relationships or activities that could appear to
have influenced the submitted work. This does not
alter our adherence to PLOS ONE policies on
sharing data and materials. There are no patents,
products in development or marketed products
associated with this research to declare.

However, there was a significant time-varying effect between anti-coagulant regimen on all-cause mortality, with DOACs showing better survival. This is a key methodological observation for future follow-up studies, and reassuring for patients and health care professionals for longer duration of therapy

## Introduction

Atrial fibrillation (AF), for all but low risk patients, is managed using anticoagulants, either direct oral anticoagulants (DOACs) or warfarin, to reduce the risk of stroke [1, 2]. DOACs are increasingly prescribed instead of warfarin in routine clinical practice, due to their fixed dosing, rapid onset, fewer dietary and drug interactions and no requirement of haematological monitoring [3–6]. However, an increased risk of mortality with the use of DOACs compared with warfarin has been reported in a recent large observational study in primary care [7].

Longer term data on stroke and mortality with DOAC use are important because meta-analyses of randomised controlled trials (RCTs) and subsequent studies are not clear about their relative safety. Meta-analysis of DOAC RCTs have shown either non-inferiority or benefit compared to warfarin therapy in the prevention of stroke in patients with non-valvular AF [2, 8–11]. However, many RCTs assessed an individual DOAC drug and importantly had limited follow-up periods of up to three years [12]. In subsequent prospective and retrospective real world studies, some over a slightly longer follow-up period, there were mixed results with respect to a greater reduction in all-cause mortality compared to ischaemic stroke (IS) [13–15].

The aims of the current study to evaluate with a longer duration of follow-up for any differences between DOAC and warfarin use in newly diagnosed cases of AF on stroke and all-cause mortality. Additionally, to highlight methodological issues in assessing outcomes over a longer follow-up period, we flagged stroke and all-cause mortality and as competing outcomes.

## Methods

### Overview

We identified 12,619 patients with incident AF between 1st January 2008 - 31st July 2019 in the Oxford Royal College of General Practitioners (RCGP) Research and Surveillance Centre (RSC) network database, a nationally representative sample of 3.5 million people [16]. We followed up cases treated continuously with a single anticoagulant treatment type (either DOAC or warfarin), until an event of interest occurred. Drug choice were made according to local guidelines or NICE recommendations. The primary outcome event of stroke or all-cause mortality was treated as a competing risk in order to avoid potential biasing of risk estimates when mortality is treated as a censoring event [17]. We utilised an incident user design in order to reduce confounding, in particular survivor bias [18].

### Population

The RCGP RSC primary care network is one Europe's oldest sentinel systems collecting data from its member general practices for over 50 years [19]. Historically it has largely been involved in the monitoring of infectious disease [19, 20] but since 2015 has been involved in a wider range of research including AF, [21] diabetes, [22] and cancer [23]. UK primary care lends itself to this type of research because: (1) It is a registration-based system, each citizen

registers with a single practice, (2) Individuals have a unique healthcare number (NHS number) that links to a "deaths and leavers table" that ensures reliability of years of exposure and all-cause mortality, (3) Computerised medical record (CMR) systems have been in place since the 1990s, [24] and (4) Prescribing is electronic, ensuring prescription numbers are accurate and it is therefore possible to measure persistence [25].

## Case ascertainment

We used clinical codes in primary care CMR systems, in the UK Read codes, [26] to identify cases of AF. Whilst key conditions such as AF have been recorded well for some years, [27] pay-for-performance (P4P) for chronic disease management, in place since 2004, have further raised data quality [28]. We excluded patients (see S1 Fig in S1 File) who had had a previous stroke and those who were not on British National Formulary recommended dose of DOAC to reduce the impact of indication bias [29]. Censoring also took place if patients de-registered from a practice or the study period ended.

**Exposures and outcomes.**   Exposures were to continuous anticoagulant prescription for the first time after receiving an AF diagnosis. Almost all anticoagulant prescriptions in the UK are issued from primary care and are, therefore, captured by the RCGP RSC database. Sometimes an anticoagulant is started in hospital; we measured exposure from the first GP prescription. We excluded patients (see S1 Fig in S1 File) who had an interval of greater than 90 days between anticoagulant prescriptions,

Outcomes were the first record of stroke and all-cause mortality, using previously published Read codes [30, 31]. Stroke was included regardless of aetiology, an approach used in other studies, [21, 32, 33] Study participants were followed for stroke and all-cause mortality up to 31 Jul 2019.

We found examples of patients receiving warfarin anti-coagulation with follow-up times exceeding seven years, longer than the DOAC cohort, for whom no examples exceed seven years of follow-up. We therefore truncated event times at seven years of follow-up, censoring events in the warfarin group that occurred after this length of follow-up.

## Covariates

We included in our study clinical, variables likely to be used as indicators in prescribing anticoagulants: age-band, gender, and deprivation reporting index of multiple deprivation (IMD) quintile. IMD is a national measure of socioeconomic status which can be derived at individual level from first part of postcode, we divided IMD into five quintiles where Q1 is the most and Q5 the least deprived. We included comorbities from the CHADVASC score and we counted the number of such risks into a cumulative score. We also adjusted for comorbidities that form part of the stroke risk score (CHA$_2$DS$_2$-VASc), [34] namely heart failure, hypertension, stroke or transient ischaemic attack, myocardial infarction or peripheral vascular disease at baseline. We categorized this variable by the number of such comorbidities into low ($< = 1$), mild (2–3) and high ($>4$) in our model. We categorised smoking into current smoker, ex-smoker or never smoker.

To control for potential confounding by year of entry study, we included a binary covariate indicating year of entry into the study (before and including 2014 and after 2014) in all multivariate analyses. 2014 was also the year that national guidance was published and may have affected the quality of prescribing [35]. Uptake of this guidance was reported (using P4P data about AF management) to be around 94%, for stroke risk assessment, and for those at risk anticoagulants being offered to 78% in 2017 rising to 86% in 2019 [36].

Finally, we reported renal function, using estimated glomerular filtration rate (eGFR). Whilst creatinine clearance is the recommended measure, [37] this is rarely calculated in primary care records and contemporary eGFR data was available for nearly all patients.

## Statistical methods

We studied the influence of anti-coagulation regimen by evaluating the cause-specific hazard ratio and the subdistribution hazard ratios of both events. Additionally, we estimated cumulative incidence for both events by direct regression, utilising the inverse of the probability of censoring weights (IPCW) method [15] with time-varying effects [38].

As recommended, [39] we report and interpret both cause specific and subdistribution analyses.

The cause specific hazard ratio (CSHR) is often interpreted as estimating aetiological association, estimating associations between covariates and the rate at which events occur in those subjects who are event-free. The cause-specific hazard ratio can be interpreted as a rate ratio. Cox proportional hazards models are employed to estimate such hazards for both events.

The subdistribution hazard ratio, evaluated by the Fine-Gray (FG) methodology, may be thought of as a measure of prognostic association, summarising predictive relationships. In one interpretation, the exponentiated regression coefficient from a subdistribution hazards model indicates the relative effect of a covariate on the instantaneous rate of occurrence in subjects who are either event-free or who have experienced a competing event. This may be an unpalatable interpretation for many as it includes subjects who have experienced the competing event and are unable to suffer the primary event. Their inclusion in the risk set after the competing event is therefore immortal time [40]. In short, it is not possible to interpret a subdistribution hazard as an epidemiological rate.

Cumulative incidence models were tested by a Kolmogorov-Smirnov type test-statistic and a Cramer von Mises type test-statistic (see S2 Table in S1 File) as well as by inspection of the Schoenfeld residuals. Such tests revealed time-varying effects for the anti-coagulation regimen for the cause-specific hazard and sub-distribution hazards analysis with respect to all-cause mortality. Despite the time-varying nature of some covariates and therefore the non-proportionality of hazards we report the CSHR and Fine-Gray analysis but interpret the hazards as time-averaged effects [40].

We carried out a sensitivity analysis on a propensity score matched (1–1) cohort after multiply imputing missing covariate values by the chained equations method. In addition to the covariates above, we included ethnicity and urban-rural and matched on these characteristics (likely to be associated with anti-coagulation prescription) using a propensity score derived from a multivariate model to perform survival analysis on the full cohort (i.e. including those with missing demographic status variables)

All analyses employed the statistical software R, version 3.5.3, additionally using the R libraries: *survival*, version 3.1–8, *cmprsk*, version 2.2–9 for estimating cause-specific hazard ratios (for right censored data and large samples), and *riskRegression* version 2019.11.3 for the subdistribution hazards estimation and the binomial cumulative incidence regression. In the sensitivity analysis, we used the *mice* library, version 3.7.0, for the imputation and the *MatchIt* library, version 3.0.2 for the propensity score matching.

## Ethical considerations

Study approval was granted by the Research Committee of the RCGP RSC. The study did not meet the requirements for formal ethics board review as defined using the NHS Health Research Authority research decision tool (**http://www.hra-decisiontools.org.uk/research/**).

The study was conducted in line with the Reporting of studies conducted using observational routinely collected data (RECORD) guidelines; [41] the cohort study diagram is included in S1 Fig of **S1 File**

## Results

### Baseline characteristics of study cohort

The incidence of AF slowly increased over the study period from 2.11 per 1,000 in 2008, to 2.99 in 2018. The incidence was consistently higher in men than women, the overall incidence rates for the period were 2.98 and 2.55/1,000 respectively. Men were generally diagnosed a decade younger (mean age 67.3) than women (mean age 73 years) over the observation period (The baseline characteristics of the study cohort with atrial fibrillation treated with either Warfarin or DOAC are shown in Table 1.

### Unadjusted rates of stroke and all-cause mortality

The crude incidence rates for stroke and all-cause mortality were 0.59 (0.52–0.67) and 4.39 (4.2–4.6) per 100 person years, respectively (Table 2, Fig 1). There were no differences in the crude incident rates of stroke between warfarin 0.59 (0.51–0.69) and DOAC 0.58 (0.44–0.75)

**Table 1. Baseline characteristics of study cohort (n, %) showing the probability of any differences between event free, stroke and all-cause mortality groups between those exposed to Warfarin or DOACs.**

| | Status | Warfarin (n = 7451) | | | DOAC (n = 5168) | | | Difference in proportion (p) comparing columns | | |
| --- | --- | --- | --- | --- | --- | --- | --- | --- | --- | --- |
| | | Event Free | stroke | All-cause mortality | Event Free | stroke | All-cause mortality | | | |
| **Sex** | Female | 2561 (41.90) | 70 (45.20) | 493 (41.30) | 2088 (44.20) | 31 (54.40) | 178 (46.20) | | | |
| | Male | 3542 (58.00) | 85 (54.80) | 700 (58.70) | 2638 (55.80) | 26 (45.60) | 207 (53.80) | <0.01 | 0.30 | 0.10 |
| **Age Band** | < = 65 years | 1148 (18.80) | 14 (9.00) | 72 (6.0) | 750 (15.90) | 4 (7.00) | 25 (6.50) | | | |
| | 65–75 years | 2025 (33.20) | 40 (25.80) | 269 (22.50) | 1756 (37.20) | 14 (24.60) | 85 (22.10) | | | |
| | >75 years | 2930 (48.10) | 101 (65.20) | 852 (71.40) | 2220 (46.90) | 39 (68.40) | 275 (71.40) | <0.00 | 0.86 | 0.94 |
| **IMD Quintile** (Q1 Least deprived) (Q5 Most deprived) | Q1 | 699 (11.50) | 12 (7.70) | 140 (11.70) | 451 (9.50) | 8 (14.00) | 51 (13.20) | | | |
| | Q2 | 850 (13.90) | 19 (12.30) | 192 (16.10) | 630 (13.30) | 9 (15.80) | 53 (13.80) | | | |
| | Q3 | 1357 (22.20) | 32 (20.60) | 291 (24.40) | 1021 (21.60) | 14 (24.60) | 88 (22.90) | | | |
| | Q4 | 1557 (25.50) | 40 (25.80) | 321 (26.90) | 1306 (27.60) | 17 (29.80) | 105 (27.30) | | | |
| | Q5 | 1640 (26.90) | 52 (33.50) | 249 (20.80) | 1318 (27.90) | 9 (15.80) | 88 (22.90) | 0.00 | 0.12 | 0.66 |
| **Comorbidities** | < = 3 | 5355 (87.70) | 126 (81.30) | 987 (82.70) | 4340 (91.80) | 50 (87.70) | 335 (87.00) | | | |
| | >3 | 748 (12.30) | 29 (18.70) | 206 (17.30) | 386 (8.20) | 7 (12.30) | 50 (13.0) | <0.00 | 0.37 | 0.06 |
| **Smoking Status** | Active Smoker | 666 (10.90) | 18 (11.60) | 144 (12.10) | 412 (8.90) | 6 (10.50) | 40 (10.4) | | | |
| | Ex-Smoker | 3637 (59.60) | 94 (60.60) | 747 (62.60) | 2936 (62.10) | 32 (56.10) | 250 (64.90) | | | |
| | Never-Smoker | 1800 (29.50) | 43 (27.70) | 302 (25.30) | 1378 (29.20) | 19 (33.30) | 97 (24.70) | <0.00 | 0.73 | 0.62 |
| **Year of Study Entry** | 2014 or before | 25067 (79.70) | 995 (88.20) | 3705 (86.60) | 367 (5.60) | 15 (9.80) | 42 (8.80) | | | |
| | After 2014 | 6367 (20.30) | 133 (11.80) | 574 (13.40) | 6139 (94.40) | 138 (90.20) | 435 (91.20) | <0.00 | <0.00 | <0.00 |
| **eGFR**[*] | | 71.8 (57.30–84.70) | 65.3 (50.90–80.10) | 63.9 (49.10–79.50) | 77.6 (64.40–87.70) | 71.0 (51.90–77.60) | 70.6 (55.40–83.30) | <0.00 | 0.97 | <0.00 |

DOAC, direct oral anticoagulant; IMD, index of multiple deprivation; **eGFR,** estimated glomerular filtration rate

[*] median and inter-quartile range

**Table 2. Crude incidence rates of stroke and all-cause mortality.**

| | Event | Events | Person years at risk | Incident rates /100 person years (95% CI) |
|---|---|---|---|---|
| **Warfarin** | Stroke | 171 | 28,878.22 | 0.59 (0.51–0.69) |
| | all-cause mortality | 1317 | | 4.56 (4.33–4.81) |
| **DOAC** | Stroke | 57 | 9905.04 | 0.58 (0.44–0.75) |
| | all-cause mortality | 385 | | 3.89 (3.52,4.29) |

however there were for all-cause mortality: warfarin prescription was associated with a higher incidence 4.56 (4.33–4.81) compared with DOACs 3.89 (3.52–4.29, p<0001, Table 2, Fig 2).

## Cause specific hazard ratios: Multivariate analysis

Among subjects who have not experienced any event, multivariate analysis suggested that the instantaneous rate of occurrence of stroke was associated with age over 75 years, CSHR 2.41 (1.44–4.00, p<0.001), eGFR, CSHR 0.61(0.41–4.00, p = 0.02) and five or more $CHA_2DS_2$-VASc listed comorbidities CSHR 2.32 (1.27–4.20, p = 0.01), Table 3, S2 Table in S1 File

The all-cause mortality rate was similarly associated with five or more $CHA_2DS_2$-VASc listed comorbidities CSHR 1.34 (1.05–1.71, p = 0.02), with age over 75years CSHR 4.12 (3.32–5.1, p<0.001) as well as age between 65 and 75 years CSHR 1.77 (1.41–2.22, p<0.001). Further instantaneous occurrence was associated with year of entry after 2014 CSHR 1.40 (1.22–1.61, p<0.001) and being in deprivation quintile 5 (the least deprived) CSHR 0.65 (0.54–0.77, p<0.001) in addition to never have smoked CSHR 0.65 (0.55–0.78, p<0.001), Table 4,

## Comparison between subdistribution hazards and cause-specific hazard ratios

There was no difference in the cause-specific hazard ratio for stroke 1.08 (0.72–1.63, *p = 0.69*) or in all-cause mortality 0.93 (0.81–1.08, *p = 0.37*); comparing the warfarin (reference) group

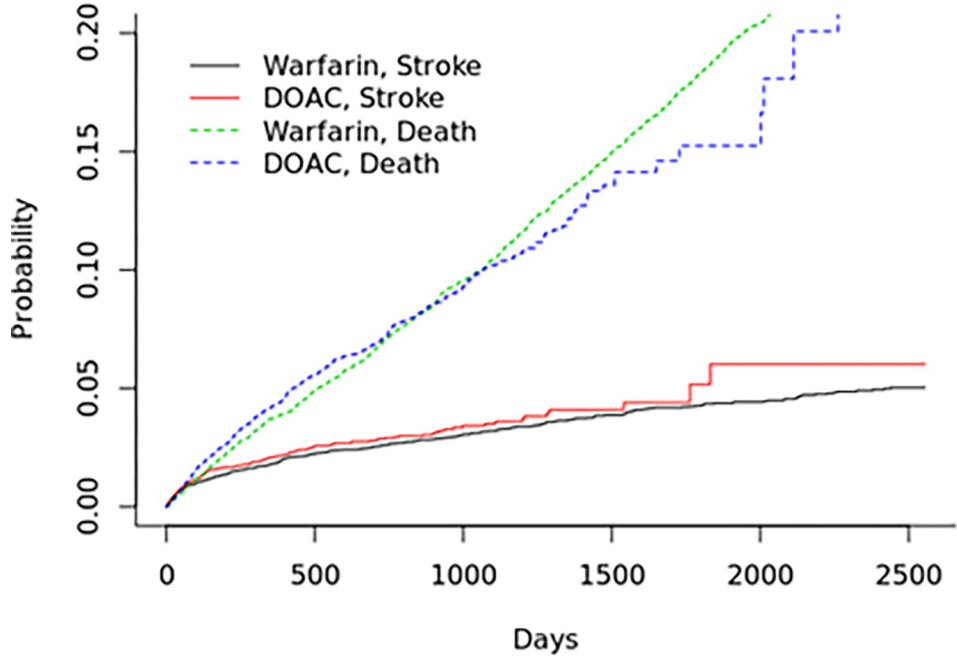

**Fig 1. Unadjusted cumulative incidence of stroke and all-cause mortality.**

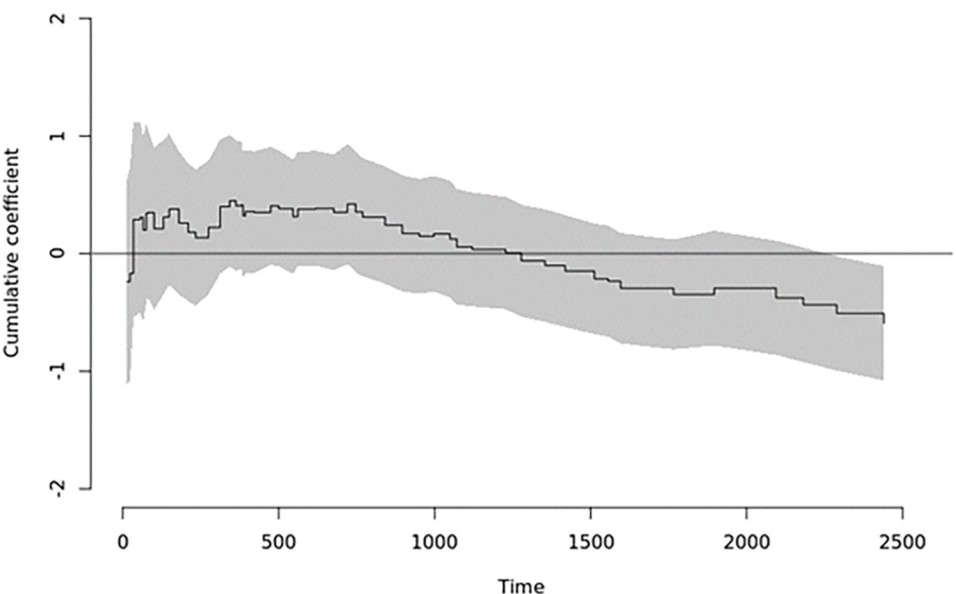

**Fig 2. Unadjusted cumulative incidence of stroke and all-cause mortality by type of anti-coagulant.** Overall test of curve separation p<0.0001 for both outcomes.

with the DOAC group. A Fine-Gray analysis produced very similar adjusted hazard ratios of 1.07 (0.71–1.6 p = 0.75) for stroke and 0.93 (0.8–1.07, p = 0.3) for mortality. These initial results suggest no difference between anti-coagulation by warfarin or by DOACs from a time averaging approach. However, whilst there was no time-varying effect for stroke, the proportional hazards assumption was not violated ($p = 0.98$, using Schoenfeld residuals), there was for mortality ($p<0.01$, Table 5).

**Table 3. Multivariate analysis of the cause-specific hazard ratio (CSHR) for stroke.**

| Variable | | Ref | HR | 95% CI | | p |
|---|---|---|---|---|---|---|
| **Anticoagulation type** | *DOAC* | Warfarin | 1.08 | 0.72 | 1.63 | 0.67 |
| **Gender** | *Male* | Female | 0.86 | **0.65** | 1.13 | 0.28 |
| **Age band** | *>65 -< = 75yrs* | ≤65yrs | 1.37 | 0.81 | 2.35 | 0.25 |
| | *>75yrs* | | 2.41 | 1.44 | 4.00 | <0.001 |
| **IMD Quintile** | *Q2* | Q1 | 1.07 | 0.61 | 1.89 | 0.82 |
| | *Q3* | | 1.08 | 0.64 | 1.83 | 0.77 |
| | *Q4* | | 1.17 | 0.70 | 1.95 | 0.55 |
| | *Q5* | | 1.20 | 0.72 | 1.99 | 0.48 |
| **log GFR** | | | 0.61 | 0.41 | 0.92 | 0.02 |
| **Comorbidities** | *2–4* | ≤1 | 1.10 | 0.77 | 1.57 | 0.60 |
| **(from CHA2DS2-VASc)** | *> = 5* | | 2.32 | 1.27 | 4.20 | 0.01 |
| **Baseline smoking status** | *Ex-smoker* | Smoker | 0.77 | 0.50 | 1.20 | 0.29 |
| | *Never* | | 0.78 | 0.48 | 1.25 | 0.30 |
| **Year of Entry** | *after 2014* | ≤2014 | 0.91 | 0.62 | 1.34 | 0.64 |

Ref, reference; HR, hazard ratio; CI, confidence interval; IMD, index of multiple deprivation; GFR, glomerular filtration rate; CHA2DS2-VASc, stroke risk score

**Table 4. Multivariate analysis of the cause-specific hazard ratio (CSHR) for all-cause mortality, interpreted as time-averaged effects.**

| Variable | | Ref | HR | 95% CI | | p |
|---|---|---|---|---|---|---|
| Anticoagulation type | DOAC | warfarin | 0.93 | 0.81 | 1.08 | 0.37 |
| Gender | Male | Female | 1.10 | 0.99 | 1.22 | 0.07 |
| Age band | >65 -< = 75yrs | ≤65yrs | 1.77 | 1.41 | 2.22 | <0.00 |
| | >75yrs | | 4.12 | 3.32 | 5.10 | <0.00 |
| IMD Quintile | Q2 | Q1 | 0.96 | 0.79 | 1.15 | 0.60 |
| | Q3 | | 0.88 | 0.74 | 1.05 | 0.16 |
| | Q4 | | 0.87 | 0.73 | 1.03 | 0.11 |
| | Q5 | | 0.65 | 0.54 | 0.77 | <0.00 |
| log GFR | | | 0.52 | 0.45 | 0.60 | <0.00 |
| Comorbidities | 2–4 | ≤1 | 0.96 | 0.85 | 1.10 | 0.55 |
| (from CHA2DS2-VASc) | > = 5 | | 1.34 | 1.05 | 1.71 | 0.02 |
| Baseline smoking status | Ex-smoker | Smoker | 0.79 | 0.67 | 0.92 | 0.00 |
| | Never | | 0.65 | 0.55 | 0.78 | <0.00 |
| Year of Entry | after 2014 | ≤2014 | 1.40 | 1.22 | 1.61 | <0.00 |

Ref, reference; HR, hazard ratio; CI, confidence interval; IMD, index of multiple deprivation; GFR, glomerular filtration rate; CHA2DS2-VASc, stroke risk score

## Risk-regression to elucidate the time varying effect

The time-varying nature of the exposure was explored by investigation of the covariate effect by risk regression (using a proportional link function), estimating sub-distribution hazards ratios with time-varying effects for anti-coagulation regimen. This allowed us to plot the time-varying effect of the anti-coagulation regime. The time-varying effect is non-significant for ischaemic stroke (Fig 3).

The effect of anti-coagulation regimen on mortality was found to be significantly time-varying: in early follow-up, around 67 days, a significantly elevated cumulative risk of mortality was present in the cohort of people prescribed a DOAC rather than warfarin. Subsequently the DOAC cumulative risk decreases until 1,537 days of follow-up when there is a significant decrease in the risk of all-cause mortality in the DOAC group: coefficient estimate: -0.23 (-0.38, -0.01, p = 0.01, Fig 4); this decreased risk persists for the follow-up period investigated in this study. In S3 Table in S1 File, we show subgroup examples.

For completeness we note that in early follow-up there is a statistically significantly elevated cumulative incidence of all-cause mortality in the DOAC group that lasts until around 1220 days (see S3 Table in S1 File). By 1537 days this coefficient estimate has become negative and decreases monotonically until end of the follow-up period.

## Sensitivity analysis

The sensitivity analysis (see S1 Table in S1 File) supports the main findings: no statistically significant differences between anti-coagulation regimen and the risk of stroke and all-cause mortality are found in the cause-specific hazard ratios and the cumulative incidence

**Table 5. Summary of multivariate adjusted cause-specific hazards and the subdistribution hazards for comparison.**

| | Stroke | 95% CI | All-cause mortality | 95% CI |
|---|---|---|---|---|
| CSHR: DOAC -v-warfarin | 1.08 | 0.72–1.63 | 0.93 | 0.81–1.08 |
| Sub-distribution HR: DOAC -v- warfarin | 1.07 | 0.71–1.60 | 0.93 | 0.80–1.08 |

CI, confidence interval; CSHR, cause specific hazard ratio; DOAC, direct oral anticoagulant; IS, ischaemic stroke

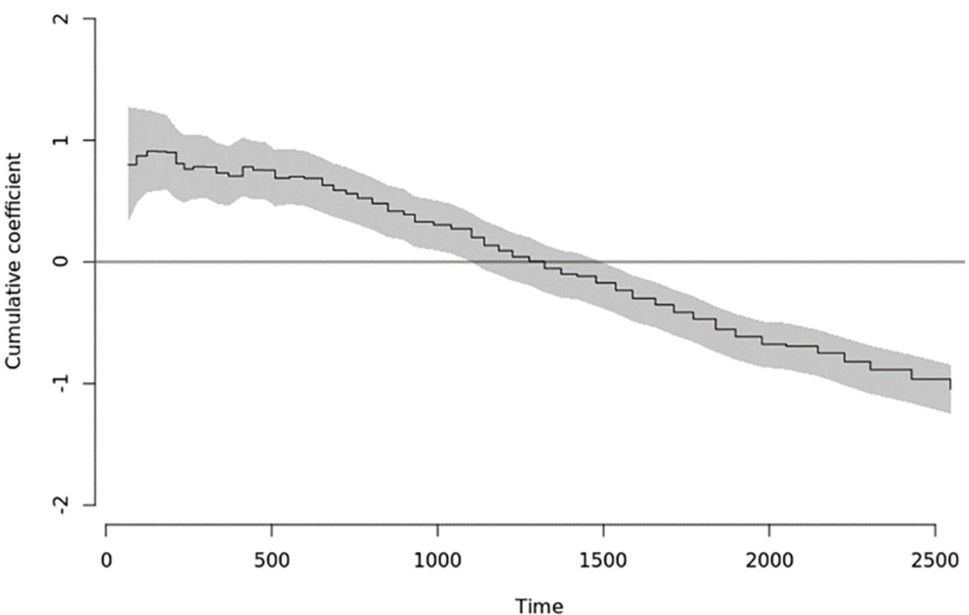

**Fig 3. Time-Varying effect of anti-coagulation regimen on sub-distribution hazards for stroke.**

regression. Furthermore, tests reveal that the proportional hazards assumption is violated for both mortality models with evidence of the time-varying nature of anti-coagulation group present in the matched analysis.

## Discussion

### Principal findings

This real world study, showed no difference between DOAC and warfarin treatment with respect to stroke, but did show a reduction in all-cause mortality with DOACs, with a key

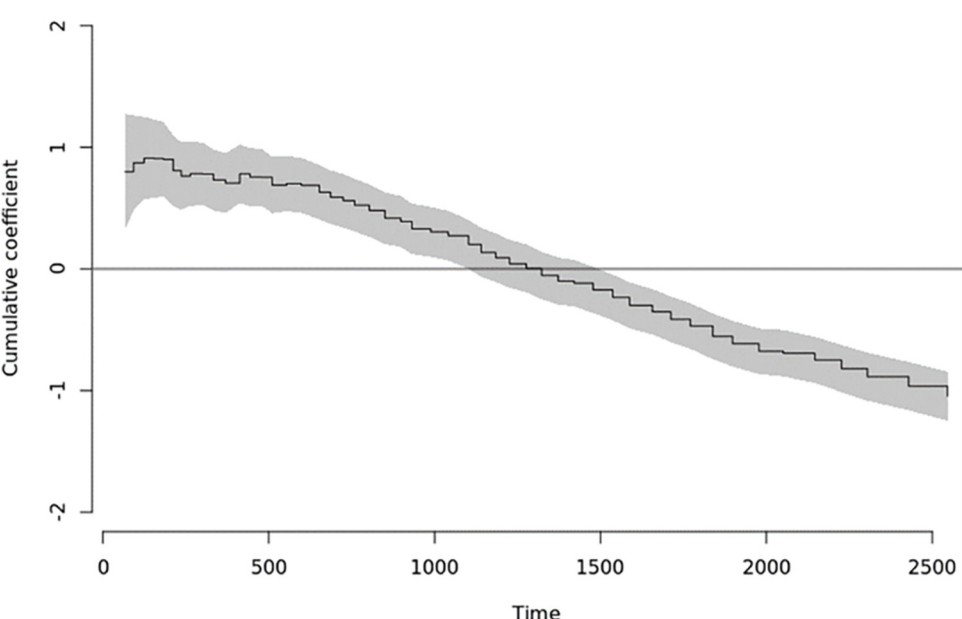

**Fig 4. Time-varying effect of anti-coagulation regime on sub-distribution hazards for all-cause mortality.**

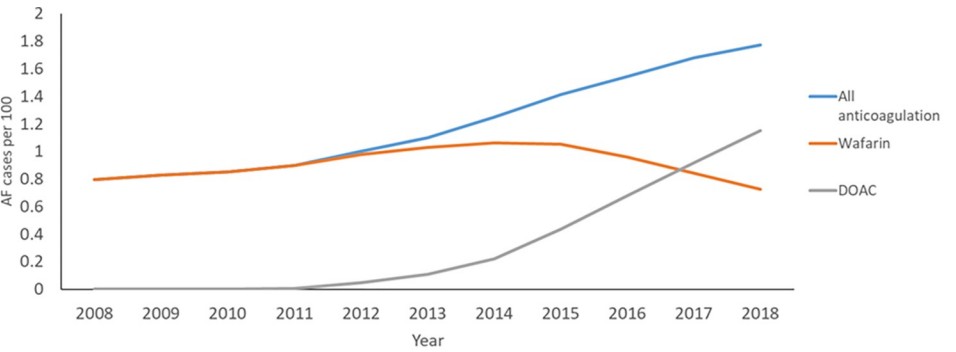

**Fig 5. Change in use of Warfarin and DOACs over time, financial incentives to encourage increased detection in the management of AF introduced in 2014.**

observation of longer follow-up compared to previous RCTs and real world studies. Importantly, the difference in favour of DOAC benefit only emerged at 1,537 days (around 4.2 years) into treatment. Given that the mean age of diagnosis of AF in men was 67 years and for women 73 years, these long-term benefits are likely to be of clinical significance and highlights the importance of staying on treatment for an extended period.

These finding are reassuring for both patients and prescribers, as the largest and previous large real world study, albeit over a shorter follow-up, had indicated an increase in all-cause mortality with DOAC use [7].

The DOAC group was associated with elevated mortality in early follow-up. However, over time, this effect significantly lowered over the warfarin group until end of follow-up at seven years. This may have been due to a 'learning curve' for general practice in using this new group of medicines (DOACs). We should interpret this as a population effect on the cumulative incidence rather than an individual risk effect and therefore is possibly due to a selection bias based on individual frailties, with the higher mortality in higher risk patients in the DOAC group which leaves leave lower risk patients, resulting in a decreasing relative cumulative incidence.

Our approach has highlighted the importance of competing risks when analysing prospective data with interlinked or interdependent clinical endpoints.

The lack of evidence of treatment effect on the cumulative incidence of stroke between people prescribed DOAC and warfarin was further underlined by significant variation over time in the exposure effect on the cumulative incidence all-cause mortality

## Strengths and limitations

The Oxford RCGP RSC network comprises a large nationally representative sample of people attending general practice throughout England. UK general practice lends itself to this type of research because it is a registration based system, providing an accurate denominator. General Practitioners have been recording data about AF for many years [42] with a high level of data completeness, [16] which has enabled the network to be an important resource for real world evidence-based research [43]. However, there is an element of selection bias since practices volunteered to join the network, with a marginal increase in affluent areas than the national population as a whole. Practices within the network have access to dashboard to improve data quality and the quality of care [44].

Financial incentives provided in 2014 in English general practice maybe important and a useful tool for other health systems wishing to promote more recognition and anticoagulation

in AF [36, 37]. This change in anticoagulant use enabled us to compare DOACs and warfarin (Fig 5).

A strength of this study was the focus on a rigorous methodological approach with regards to appropriate statistical analyses. The cause-specific hazards and the subdistribution hazards are summarised for comparison in Table 5. The concordance between the adjusted cause-specific and sub-distribution hazards ratios is to be expected since the cumulative incidence of stroke is small compared to the incidence of all-cause mortality [27].

We found a non-significant exposure effect on the cumulative incidence of stroke, which further underlined the lack of evidence in variation in time to stroke between people prescribed DOAC and warfarin. However, there we a significant variation over time in the exposure effect on the cumulative incidence all-cause mortality.

Possible selection bias may also limit our study as some practices may use one of the medicines more than another according to specific clinical characteristics, limitations or risks. We did not utilise the CHA2DS2VASc score in our analysis due to the potential problem of calculator implementation on GP computer system already highlighted in previous publications [45, 46]. In future studies we could consider data linkage to hospital and registry data be able to more precisely differentiate ischaemic from haemorrhagic stroke. This may also allow us to identify patients who have suffered haemorrhages, and those that required hospital admission.

### Further research

An additional study over a longer period of time and using the enlarged Oxford RCG RSC network would be supported by the enthusiasm among general practice colleagues during-pandemic research which has grown our network to over 1,600 practices [47]. Further research should explore precise stratification of the population by CHADVASC score, risks of haemorrhage identify causes of mortality and level of frailty and evaluate if there is any differential effect between drugs in class It is also possible that with longer follow-up there might be an emergent difference in stroke.

### Conclusions

In this real world study with a longer follow-up compared to previous studies, we found no difference between DOAC and warfarin treatment for atrial fibrillation with respect to stroke reduction. Taking into account that stroke and mortality are competing endpoints, we found a significant time-varying effect for specific anti-coagulation drug on all-cause mortality. People prescribed DOACs had elevated mortality in early follow-up, however over time, this was significantly lowered compared with warfarin right through until end of follow-up at seven years. This is a key methodological observation for future follow-up studies, but additionally reassuring from a therapeutic viewpoint for patients and health care professionals for long duration of therapy

### Supporting information

**S1 File.**
(DOCX)

### Acknowledgments

Patients and member practices of the Oxford RCGP RSC sentinel network for allowing data sharing. EMIS, TPP, In-Practice Systems and Apollo Medical Solutions for facilitating pseudonymised data extracts.

## Author Contributions

**Conceptualization:** Simon de Lusignan, F. D. Richard Hobbs, Harshana Liyanage, Christian Heiss, Michael Feher, Mark P. Joy.

**Data curation:** Harshana Liyanage, Julian Sherlock.

**Formal analysis:** Harshana Liyanage, Mark P. Joy.

**Methodology:** Simon de Lusignan, Julian Sherlock, Filipa Ferreira, Manasa Tripathy, Christian Heiss, Michael Feher, Mark P. Joy.

**Project administration:** Filipa Ferreira, Manasa Tripathy.

**Writing – original draft:** Simon de Lusignan, F. D. Richard Hobbs.

**Writing – review & editing:** Simon de Lusignan, F. D. Richard Hobbs, Harshana Liyanage, Julian Sherlock, Filipa Ferreira, Manasa Tripathy, Christian Heiss, Michael Feher, Mark P. Joy.

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
