## [Decision Letter · Decision Letter 0]

18 May 2021

PONE-D-21-07035

Long term follow up of direct oral anticoagulants and warfarin therapy on stroke, with all - cause mortality as a competing risk, in people with atrial fibrillation: sentinel network database study.

PLOS ONE

Dear Dr. de Lusignan,

Thank you for submitting your manuscript to PLOS ONE. After careful consideration, we feel that it has merit but does not fully meet PLOS ONE’s publication criteria as it currently stands. Therefore, we invite you to submit a revised version of the manuscript that addresses the points raised during the review process.

We look forward to receiving your revised manuscript.

Kind regards,

Michael Nagler, M.D., Ph.D., MSc

Academic Editor

PLOS ONE

Journal Requirements:

2. In ethics statement in the manuscript and in the online submission form, please provide additional information about the patient records/samples used in your retrospective study. Specifically, please ensure that you have discussed whether all data/samples were fully anonymized before you accessed them and/or whether the IRB or ethics committee waived the requirement for informed consent. If patients provided informed written consent to have data/samples from their medical records used in research, please include this information.

3.Thank you for stating the following in the Acknowledgments Section of your manuscript:

"FDRH acknowledges part-funding from the National Institute for Health Research (NIHR)

428 School for Primary Care Research, the NIHR Collaboration for Leadership in Health Research and

429 Care (CLARHC) Oxford, the NIHR Oxford Biomedical Research Centre (BRC, UHT), and the NIHR

430 Oxford Medtech and In-Vitro Diagnostics Co-operative (MIC)."

 "The work was supported by an unconditional grant to SdeL. Grant number DSJP3700 by Daiichi Sankyo Limited. URL: https://www.daiichi-sankyo.co.uk.The funder had no role in  study design, data collection and analysis, decision to publish, or preparation of the manuscript"

4.Thank you for stating the following in the Competing Interests section:

"I have read the journal's policy and the author of this manuscript has the following competing interest: Simon de Lusignan is the Director of the Oxford RCGP RSC and has received funding through his University for studies from Eli Lilly, Astra-Zeneca, Sanofi, GSK, Seqirus and Takeda; and been member of advisory boards for influenza for Seqirus and Sanofi. FDRH has received occasional fees from Bayer and Boehringer Ingelheim for speaking or consulting on atrial fibrillation related stroke risk. All other authors have declared no financial relationships with any organisations that might have an interest in the submitted work in the previous three years, no other relationships or activities that could appear to have influenced the submitted work."

6.We note that you have indicated that data from this study are available upon request. PLOS only allows data to be available upon request if there are legal or ethical restrictions on sharing data publicly. For more information on unacceptable data access restrictions, please see http://journals.plos.org/plosone/s/data-availability#loc-unacceptable-data-access-restrictions.

Reviewers' comments:

Reviewer's Responses to Questions

**Comments to the Author**

1. Is the manuscript technically sound, and do the data support the conclusions?

Reviewer #1: Partly

Reviewer #2: Yes

2. Has the statistical analysis been performed appropriately and rigorously? 

Reviewer #1: Yes

Reviewer #2: Yes

3. Have the authors made all data underlying the findings in their manuscript fully available?

Reviewer #1: No

Reviewer #2: Yes

4. Is the manuscript presented in an intelligible fashion and written in standard English?

Reviewer #1: No

Reviewer #2: Yes

5. Review Comments to the Author

Reviewer #1: The authors conducted a large study investigating mortality and stroke in patients with newly diagnosed atrial fibrillation treated either with DOAC or warfarin. Routinely obtained data of the RCGP database were used. This is an interesting study. I have three major comments which should be addressed to improve the manuscript before publication.

1. While reading the manuscript, it remains unclear why the study was conducted and what the results mean (in the context of previous literature). The clinical problem or the unsolved scientific question leading to this study is not clearly stated. Thus, it is also not clear how to interpret the results.

2. The authors used elaborated statistical techniques to analyze the data. However, it is not explained how these techniques help to answer the research questions raised. This issue increases the problem of how to interpret the results of this study. Besides, please reword the results (results section, table headings, figure legends) in a way that the results answer the research question.

3. The authors used a database of routinely obtained data to do the analysis. A large number of pitfalls are possible in this kind of study that might lead to biased results or wrong interpretations. The authors already spend some efforts to convince the reader that the results are valid. But I strongly believe that the authors must give more details how the outcomes (stroke, death), and co-variates are recorded and how completeness of these measurements is ensured. Besides, incomplete data are a major risk of these kind of studies and the authors must elaborate on this in the limitations part of the manuscript.

4. The authors did not discuss the results in context of previous literature. What did other studies find? Why are the results different? It is the methodology, I guess. Please add a comprehensive paragraph on this issue referring to all key studies comparing the risk of stroke and mortality between DOAC and Warfarin in patients with atrial fibrillation.

Some detailed comments as examples:

1. Title: What is a ‘sentinel network database study’? Please describe early in the manuscript or even in the abstract.

2. Abstract: It appears that the abstract is not formatted according to PLOS one.

3. Abstract/ Background: Please describe the clinical problem or the unresolved scientific question that leads to the present study.

4. Abstract/ Methods: The population is not fully described. Please add more information on the inclusion criteria.

5. Abstract/ Results: You state the statistical methods and results, but it is hard to follow what this means in terms of the research question. I suggest rewording it in a way that a reader can follow who is less familiar with statistical concepts.

6. Abstract/ Results: what does long-term mean? Please state the observation time.

7. Abstract/ Results: Patient characteristics are not given. Please state at least age and sex.

8. Abstract/ Results: Outcomes (incidence rates of stroke, death are not given)

9. Abstract/Conclusions: You did not find a difference in hazards of stroke between DOAC and Warfarin. This is in contrast to a number of previous studies. Why? It’s the methodology I guess. Please clarify this already in the abstract.

Reviewer #2: In this study, the authors analyzed a retrospective cohort of 12619 primary care patients with a first-time diagnosis of atrial fibrillation who were treated with either DOAC or warfarin and performed a competing risk analysis between stroke and all-cause mortality. They conclude that there is no significant difference of hazard between anticoagulants for stroke but all-cause mortality DOACs performed better. While this study is certainly of interest some points need to be addressed before publication:

Major Comments:

(Page 4, Line 3): It is appreciated that the authors thoroughly describe all analysis, however, the background section of the abstract is lacking a sentence about the context of the study (Why did you perform this study?)

(Page 5, Lines 54-57): While reading this part of the introduction, I did not clearly understand the authors' research question. The authors might want to change the wording here

(Page 6,) Thank you for describing the database in such detail. However, some information would still improve the understandability. Are comorbidities entered in the database by the physician or derived from the UK read codes? Is the smoking status also recorded by the physician?

(Page 8): While the statistical analysis is described in-depth, I still wonder how you handled missing data for the primary analysis. You write that you performed a sensitivity analysis with multiple imputations, but did you perform a complete-case analysis for the primary analysis? Please consider describing this more in detail.

(Page 10): The authors might want to include the median observation time of the two groups (DOAC vs. Warfarin).

(Figure 2): I am not sure if the figure legend here is correct. The legend text suggests that there should be more than one curve (e.g. “curve separation”) but the figure just shows one curve.

Minor comments:

(Page 5, Line 54-57) Although I am not a native English speaker, the first sentence seems to miss a verb. The second sentence seems to have an “and” after all-cause mortality too much.

(Page 7): The authors might want to consider moving the flow chart from Figure S1 from the supplementary material to the main article. In my opinion, it would improve readability.

(Page 7, Line 90): A small typo. I think the “to” is too much here.

(Page 8 Lines 109-113): The authors might want to revise the wording in these sentences. It remains unclear to me.

(Table 1): You present here an expansive table. In my opinion, including the number of missing values per covariate would further improve the table. Also, the third column (“Difference in proportion”) is quite hard to understand. You might want to write in the table legend that the p-values are for the comparison of Warfarin to DOAC patients.

(Page 15, Line 238): Probably just a typo but the upper confidence interval of the eGFR is according to table 3 0.92, not 4.00.

(Figure 3 & Figure 4): It might be helpful to show the sub-distribution in one figure to make the graphs easier to compare.

6. PLOS authors have the option to publish the peer review history of their article (what does this mean?). If published, this will include your full peer review and any attached files.

Reviewer #1: **Yes: **Michael Nagler

Reviewer #2: No

---

## [Author Response · Author response to Decision Letter 0]

15 Feb 2022

Comments from Reviewer #1

1. While reading the manuscript, it remains unclear why the study was conducted and what the results mean (in the context of previous literature). The clinical problem or the unsolved scientific question leading to this study is not clearly stated. Thus, it is also not clear how to interpret the results.

Response

For clarity we have re-phrased the aims of the study, conforming to the reviewer's comments. We hope that this aids the interpretation of the statistical results. 

2. The authors used elaborated statistical techniques to analyze the data. However, it is not explained how these techniques help to answer the research questions raised. This issue increases the problem of how to interpret the results of this study. Besides, please reword the results (results section, table headings, figure legends) in a way that the results answer the research questions.

Response

A senior statistician was part of the research team and conducted an appropriate statistical analysis for a time to event analysis, with outcome stroke (and a competing risk of all-cause mortality). In the light of the updated research aim, to provide real-world evidence re stroke outcomes, we anticipate that the results will be easier to interpret.

3. The authors used a database of routinely obtained data to do the analysis. A large number of pitfalls are possible in this kind of study that might lead to biased results or wrong interpretations. The authors already spend some efforts to convince the reader that the results are valid. But I strongly believe that the authors must give more details how the outcomes (stroke, death), and co-variates are recorded and how completeness of these measurements is ensured. Besides, incomplete data are a major risk of these kind of studies and the authors must elaborate on this in the limitations part of the manuscript. 

Response

We have amended the "Case ascertainment section": We refer the reviewer to lines 88-95 in revised m/s

4. The authors did not discuss the results in context of previous literature. What did other studies find? Why are the results different? It is the methodology, I guess. Please add a comprehensive paragraph on this issue referring to all key studies comparing the risk of stroke and mortality between DOAC and Warfarin in patients with atrial fibrillation.

Response

We refer the reviewer to the m/s lines 47-54 and references 4-7. 

Reviewer #1: Detailed Comments

4. Abstract/ Methods: The population is not fully described. Please add more information on the inclusion criteria. 

Response

We would refer the reviewer to the para. 2 of the introduction (beginning line 45) for discussion around comparison with previous studies.

1. Title: What is a ‘sentinel network database study’? Please describe early in the manuscript or even in the abstract.

Response

We have updated the abstract with a brief explanation of a sentinel primary care network.

2. Abstract: It appears that the abstract is not formatted according to PLOS 

Response

As per PLOS ONE requirements we now have the sections: Title (affiliations), Abstract, Introduction, followed by the Middle section, etc. The original Methods and Conclusions section have migrated to the Middle section. We have also updated formatting.

3. Abstract/ Background: Please describe the clinical problem or the unresolved scientific question that leads to the present study

Response

We have updated the background section in the manuscript to address this issue.

4. Abstract/ Methods: The population is not fully described. Please add more information on the inclusion criteria

Response

See Supplementary Fig. S1 has been included in the main m/s.

5. Abstract/ Results: You state the statistical methods and results, but it is hard to follow what this means in terms of the research question. I suggest rewording it in a way that a reader can follow who is less familiar with statistical concepts

Response

Please see our response to Point 2 in the section preceding the Detailed comments section.

6. Abstract/ Results: what does long-term mean? Please state the observation time.- 

Response

We have written this more clearly in the m/s that the follow-up time was up to 7 years.

7. Patient characteristics are not given. Please state at least age and sex

Response

 Please refer to Table 1: Baseline characteristics of study cohort

8. Abstract/ Results: Outcomes (incidence rates of stroke, death are not given)

Response

Please refer to section: "Unadjusted rates of stroke and all-cause mortality"

9. Abstract/Conclusions: You did not find a difference in hazards of stroke between DOAC and Warfarin. This is in contrast to a number of previous studies. Why? It’s the methodology I guess. Please clarify this already in the abstract.

Response

In the Strengths and limitations section we have highlighted the possibility of selection bias re utilisation of drugs in primary care units and also the potential problem of fully recording the CHA2DS2VASc score during primary care consultations. RCTs also trial one drug. For these reasons it may be speculated that our results differ from those of RCTs.

Comments from Reviewer #2

Major Comments:

It is appreciated that the authors thoroughly describe all analysis, however, the background section of the abstract is lacking a sentence about the context of the study (Why did you perform this study?)

While reading this part of the introduction, I did not clearly understand the authors' research question. The authors might want to change the wording here.

Response

For our response to these comments we refer the reviewer to the updated abstract

Are comorbidities entered in the database by the physician or derived from the UK read codes? Is the smoking status also recorded by the physician?

READ codes are recorded by the primary care physician during all patient consultation. These codes are entered electronically during consultation and are then batch-uploaded to a central repository which then supplies the RCGP RSC database used in this study. Smoking is recorded by the physician.

(Page 8): While the statistical analysis is described in-depth, I still wonder how you handled missing data for the primary analysis. You write that you performed a sensitivity analysis with multiple imputations, but did you perform a complete-case analysis for the primary analysis? Please consider describing this more in detail.

Response 

A compete cases analysis was considered as the primary mode of analysis. We have updated the manuscript to emphasise this point.

(Page 10): The authors might want to include the median observation time of the two groups (DOAC vs. Warfarin).

Fig. 2 what's gone wrong??

Response

We have inserted the correct Figure in the updated m/s.

missing verb: 

Thank you for the comment: appropriately updated.

(Page 7): The authors might want to consider moving the flow chart from Figure S1 from the supplementary material to the main article. In my opinion, it would improve readability.

Response

Cohort diagram has been included in m/s as Figure 1 -other Fig.s have numbering updated.

(Page 7, Line 90): A small typo. I think the “to” is too much here.

Response

Deleted agreed

(Table 1): You present here an expansive table. In my opinion, including the number of missing values per covariate would further improve the table. Also, the third column (“Difference in proportion”) is quite hard to understand. You might want to write in the table legend that the p-values are for the comparison of Warfarin to DOAC patients.

Response 

We have updated the bottom legend of this Table

(Page 15, Line 238): Probably just a typo but the upper confidence interval of the eGFR is according to table 3 0.92, not 4.00.

Response

Corrected - thank you.

IMD is a UK-nationally accepted index of SES; similarly, CHA2DS2-VASc is an international index of cardiovascular risk (ref 34 in m/s).

(Figure 3 & Figure 4): It might be helpful to show the sub-distribution in one figure to make the graphs easier to compare.

Response

We have relabelled the figures and trust that this suffices

---

## [Editor Report · Decision Letter 1]

14 Mar 2022

Long term follow up of direct oral anticoagulants and warfarin therapy on stroke, with all-cause mortality as a competing risk, in people with atrial fibrillation: sentinel network database study.

PONE-D-21-07035R1

Dear Dr. de Lusignan,

We’re pleased to inform you that your manuscript has been judged scientifically suitable for publication and will be formally accepted for publication once it meets all outstanding technical requirements.

Kind regards,

Michael Nagler, M.D., Ph.D., MSc

Academic Editor

PLOS ONE
---

## [Editor Report · Acceptance letter]

23 Aug 2022

PONE-D-21-07035R1 

Long term follow up of direct oral anticoagulants and warfarin therapy on stroke, with all-cause mortality as a competing risk, in people with atrial fibrillation: sentinel network database study. 

Dear Dr. de Lusignan:

I'm pleased to inform you that your manuscript has been deemed suitable for publication in PLOS ONE. Congratulations! Your manuscript is now with our production department. 

Kind regards, 

on behalf of

Prof. Dr. Michael Nagler 

%CORR_ED_EDITOR_ROLE%

PLOS ONE